# Combinatorial Blood Platelets-Derived circRNA and mRNA Signature for Early-Stage Lung Cancer Detection

**DOI:** 10.3390/ijms24054881

**Published:** 2023-03-02

**Authors:** Silvia D’Ambrosi, Stavros Giannoukakos, Mafalda Antunes-Ferreira, Carlos Pedraz-Valdunciel, Jillian W. P. Bracht, Nicolas Potie, Ana Gimenez-Capitan, Michael Hackenberg, Alberto Fernandez Hilario, Miguel A. Molina-Vila, Rafael Rosell, Thomas Würdinger, Danijela Koppers-Lalic

**Affiliations:** 1Department of Neurosurgery, Cancer Center Amsterdam, Amsterdam University Medical Center, Vrije Universiteit Amsterdam, 1081 HV Amsterdam, The Netherlands; 2Department of Genetics, Faculty of Science, University of Granada, 18071 Granada, Spain; 3Bioinformatics Laboratory, Biomedical Research Centre (CIBM), PTS, 18100 Granada, Spain; 4Department of Biochemistry, Molecular Biology and Biomedicine, Universitat Autónoma de Barcelona (UAB), 08193 Cerdanyola, Spain; 5Germans Trias i Pujol Health Sciences Institute and Hospital (IGTP), 08916 Barcelona, Spain; 6Pangaea Oncology—Laboratory of Oncology, Quirón Dexeus University Hospital, Sabino Arana 5-19, 08028 Barcelona, Spain; 7Andalusian Research Institute in Data Science and Computational Intelligence, University of Granada, 18071 Granada, Spain; 8Brain Tumor Center Amsterdam, Cancer Center Amsterdam, Amsterdam University Medical Center, Vrije Universiteit Amsterdam, 1081 HV Amsterdam, The Netherlands; 9Mathematical Institute, Leiden University, 2333 CA Leiden, The Netherlands

**Keywords:** liquid biopsy, biomarkers, circular RNA, messenger RNA, platelets, lung cancer, cancer diagnosis

## Abstract

Despite the diversity of liquid biopsy transcriptomic repertoire, numerous studies often exploit only a single RNA type signature for diagnostic biomarker potential. This frequently results in insufficient sensitivity and specificity necessary to reach diagnostic utility. Combinatorial biomarker approaches may offer a more reliable diagnosis. Here, we investigated the synergistic contributions of circRNA and mRNA signatures derived from blood platelets as biomarkers for lung cancer detection. We developed a comprehensive bioinformatics pipeline permitting an analysis of platelet-circRNA and mRNA derived from non-cancer individuals and lung cancer patients. An optimal selected signature is then used to generate the predictive classification model using machine learning algorithm. Using an individual signature of 21 circRNA and 28 mRNA, the predictive models reached an area under the curve (AUC) of 0.88 and 0.81, respectively. Importantly, combinatorial analysis including both types of RNAs resulted in an 8-target signature (6 mRNA and 2 circRNA), enhancing the differentiation of lung cancer from controls (AUC of 0.92). Additionally, we identified five biomarkers potentially specific for early-stage detection of lung cancer. Our proof-of-concept study presents the first multi-analyte-based approach for the analysis of platelets-derived biomarkers, providing a potential combinatorial diagnostic signature for lung cancer detection.

## 1. Introduction

With 1.8 million deaths per year, lung cancer remains the leading cause of cancer mortality worldwide [1]. This high mortality can be attributed to two main reasons: late diagnosis and the inefficiency of the treatments available. Most of the patients present an advanced stage of the disease at the time of diagnosis, leading to an expected survival at 5 years below 10% [2,3,4,5]. Novel reliable, sensitive, and accurate diagnostic tests are required since early-stage identification is associated with longer life expectancy.

In recent years, liquid biopsy (LB) has been proposed as a highly promising diagnostic approach for the detection and management of cancer patients. An analysis of tumor-derived biomarkers present in human body fluids offers a minimally invasive, safe, and sensitive alternative or complementary approach to tissue biopsies. Besides the commonly used blood-based biosources and biomolecules, such as circulating tumor cells (CTCs), cell-free DNA (cfDNA), and extracellular vesicles (EVs), blood platelets have recently emerged as promising novel carriers of cancer biomarkers [6,7,8]. Platelets dynamically interact with tumor cells, which can lead to a direct and an indirect alteration of their transcriptome [9]. Changes in the RNA profile of these tumor-educated platelets (TEPs) can be used as a surrogate signature for the detection, localization, and molecular profiling of different types of cancer [10,11,12,13,14]. Furthermore, it has been established that a considerable fraction of platelets are also generated within the lung, which may position them as a more advantageous indicator of lung cancer due to the possible impact of the disease on platelet formation [15,16,17,18].

Platelet RNA repertoire includes several types of RNA families which can be potentially used as biomarkers. A first insight of the diagnostic potential of the TEPs transcriptome was described during the profiling of the platelet mRNA repertoire of metastatic lung patients and asymptomatic individuals. This study discovered that the presence of cancer results in altered spliced mRNA profiles [19]. Afterwards, the use of platelet spliced mRNA as a biomarker for the detection and classification of various tumor types has been investigated in numerous studies [10,11,20,21,22].

More recently, the expression of other types of RNAs has been found dysregulated in platelets [14,23]. In particular, human platelets are highly enriched in circular RNA (circRNA) [24]. This type of RNA is characterized by a covalent loop structure generated by a noncanonical alternative splicing process named back-splicing. Due to their high stability, abundance, and spatiotemporal specific expression, circRNA have received increasing attention for their potential role as cancer biomarkers [25]. Recently, we have provided evidence that platelet-derived circRNA profile changes in the presence of NSCLC, indicating that circRNA may hold the potential as a biomarker for liquid biopsy tests [14].

Previous studies on platelet transcriptome were based on the use of RNA-seq technology. Although RNA-seq is currently the most used methodology for genomic-based biomarker discovery, its implementation in the clinic has several limitations due to its time-consuming and elaborate library preparation protocol, the lack of standardized methods, the high cost, and complex data analysis [26].

NanoString nCounter, a platform for the high-throughput analysis of gene expression, has grown in popularity both in clinical settings and in translational research due to its fast, simple, and reliable protocol. By directly hybridizing and counting the individual targets, nCounter technology enables the multiplex analysis of signatures up to 800 genes with high reliability and reproducibility. In contrast to RNA-seq methods, nCounter RNA analysis does not require reverse transcription, amplification, nor cDNA library construction. Altogether, all these features make this system less prone to bias, leading to a more accurate quantification of the targets. Clinical tests have been developed employing nCounter technology, including the FDA-approved nCounter Prosigna test, which stratifies breast cancer subtypes and predicts recurrence risk in post-menopausal women [27,28], and the tumor inflammation signature (TIS) assay, which forecasts PD-1 checkpoint blockade and clinical response across several tumor types [29]. This platform has also been employed for the discovery of potential biomarker signatures in various types of LB biosources, including cfDNA, cell-free RNA (cfRNA), EVs (including DNA, micro RNA (miRNA) and mRNA), as well as CTCs [30,31,32,33,34,35,36]. However, the platelet transcriptome has not been explored yet through this technology for LB purposes.

Here, we present the development of a protocol for the interrogation of platelet mRNA and circRNA repertoire using NanoString nCounter technology and machine learning (ML) approaches. We applied this methodology to the platelet transcriptome obtained from lung cancer and non-cancer individuals in order to identify and evaluate the diagnostic value of each of the individual mRNA and circRNA signatures. Since a single type of biomarker may lack sensitivity and specificity for the enrichment of reliable clinical diagnostics information, we also explore a multi-analyte-based approach, using a combinatorial analysis of platelets-derived mRNA and circRNA to improve the detection of lung cancer.

## 2. Results

### 2.1. Analysis of Blood Platelet-Derived RNA Using NanoString nCounter Technology

We investigated if a direct platelet RNA analysis might provide adequate gene expression information without performing any pre-amplification step. Due to the limited amount of total RNA present in platelets, we tested different concentrations to determine the minimum amount of RNA input necessary to preserve critical gene expression information. Six different RNA concentrations (1 ng, 3 ng, 6 ng, 12 ng, 24 ng, and 48 ng) obtained from platelets of a lung cancer patient and a non-cancer individual (indicated as control) were analyzed by using the human immunology v2 panel (Appendix A) [37]. As expected, the highest total number of counts (after negative background removal) was observed using 48 ng of total RNA (Figure 1a,b, Appendix A). The number of counts decreases along with the concentration, following a linear regression model (R^2^ = 0.99, *p*-value < 0.0001, both for cancer and control samples, Figure 1a,b), suggesting that hybridization efficiency between probes and RNA remains consistent also at the lowest concentrations. Similar results were obtained when considering the average counts per transcript (R^2^ = 0.98, *p*-value < 0.0001, for control and R^2^ = 0.97, *p*-value < 0.0002 for cancer, Appendix A) confirming the previous observations.

However, we found a significant drop in the number of transcripts detected when 1 ng and 3 ng of total RNA were used compared with higher concentrations (Appendix A). Using PCA analysis, we observed that samples generated with total RNA inputs of 1 ng and 3 ng deviated from the main cluster that encompassed the other concentrations examined. This implies that the RNA profiles of samples generated with 3 ng or less of RNA input are not consistent with those acquired with higher RNA input, which could hamper subsequent gene expression analyses (Figure 1c,d).

Therefore, we conclude that a minimum concentration of 6 ng of platelet RNA without pre-amplification process is recommended for sufficient and robust transcripts expression profiles for platelet-RNA analysis with nCounter.

### 2.2. Profiling mRNA and circRNA Derived from Lung Cancer Patients and Non-Cancer Individuals Using Human Immunology V2 Panel and 78-circRNA Custom Panel

Following the protocol described in Figure 2, we investigated the potential use of platelet mRNA (using human immunology v2 panel [37]) and circRNA (with the 78-circRNA custom panel [38]) as diagnostic biomarkers. We selected a cohort of 60 platelet samples isolated from lung cancer patients (*n* = 30) and non-cancer controls (*n* = 30) equally distributed per age and gender (Table 1). Since early-stage detection is crucial for lung cancer diagnosis, we selected samples from patients with mainly early-stage (from stage IA to stage IIIA) lung cancer (*n* = 20) while the remaining samples were from patients diagnosed with metastatic tumor stage (*n* = 10). We include both asymptomatic individuals (*n* = 27) and samples from patients with confirmed benign lung nodules (*n* = 3) in the control group. Total RNA extracted from platelets was stored in RNAlater (as explained in Section 4 Materials and Methods) and checked for quality before further processing (Appendix A–F).

After subtracting the background (negative control) signal, we observed that 159 out of the 594 genes in the human immunology v2 panel were not present in any of the processed samples. A total of 402 platelets-derived mRNA were detected in both the control and cancer groups, whereas 18 transcripts were exclusively found in the control group and 15 in the lung cancer group (Figure 3a).

All the 78 circRNAs present in the custom-made panel were detected in at least one of the samples. Only three circRNAs (circNOL6, circPTPRM, circGAyS8) were exclusively detected in the lung cancer group (Figure 3b). All these three circRNAs have been previously found to be dysregulated in lung cancer [39,40,41,42].

The analysis of the average number of transcripts detected per group using the human immunology v2 panel revealed 185 ± 97 mRNAs in the control group and 218 ± 85 mRNAs in the cancer group (Figure 3c). Although the average number of transcripts is slightly higher in the cancer group than in the control, the difference is not statistically significant (ns) (Mann–Whitney’s U *p*-value > 0.05) (Figure 3c).

Out of the 78 circRNA present in the custom-made panel, an average of 54 ± 8 circRNA were detected in the cancer group and 53 ± 9 for the control group (Figure 3d). Moreover, in this case, no statistical difference between the two groups was observed (Mann–Whitney’s U *p*-value > 0.05) (Figure 3d).

### 2.3. Normalization of the Raw Counts and Differential Gene Expression between Lung Cancer Patients and Non-Cancer Individuals

mRNA raw count data (Appendix A) was evaluated prior to normalization through analytical exploratory analysis. Assessment of the unnormalized mRNA raw data analysis utilizing a PCA plot reveals no significant batch effect or clear group cluster separation (Appendix A). To prevent inaccurate normalization due to genes with low expression and background noise, we removed 314 mRNA targets (as explained in Section 4 Materials and Methods) from the analysis. Moreover, based on the interquartile range method (1.5 IQR rule), two out of sixty samples were identified as possible outliers (Appendix A). Additionally, these samples also presented aberrant values for binding density and positive control linearity; therefore, they were excluded from subsequent data processing.

Since an optimal normalization of the data is key for precise and consistent outcomes, we compared two different approaches: edgeR and DESeq2. Based on the RLE analysis, DESeq2 was found to perform better than edgeR in normalizing the mRNA data (DESeq2 R^2^ = 0.002 (Figure 4a) and edgeR R^2^ = 0.036 (Appendix A)).

Differential expression analysis between lung cancer and the control group revealed a total of 25 significantly differentially expressed mRNA (|FC| > 0.5 and *p*-adj < 0.05), of which 15 were upregulated and 10 downregulated in lung cancer patients (Figure 4b).

The circRNA raw count data (Appendix A) have been processed following the same filtering and normalization procedure as previously performed for mRNA data. The PCA plot evaluation reveals no apparent class grouping or substantial batch impact (Appendix A). Only five of the seventy-eight circRNA targets were excluded due to low expression (see Section 4 Materials and Methods). Two samples were flagged by IQR analysis as potential outliers (Appendix A). Since neither of them deviated from the main cluster in the PCA plot or showed any anomalies on the standard control metrics supplied by NanoString, both samples were kept in the dataset for further analysis. Similarly for the mRNA data, the DESeq2 package was found to obtain a more precise normalization of the data (DESeq2 R^2^ = 0.002 (Figure 4c) and edgeR R^2^ = 0.023 (Appendix A)). Differential expression analysis identified only one circRNA (circFUT8) as significantly upregulated in the lung cancer group (|FC| > 0.5 and *p*-adj < 0.05, Figure 4d). Interestingly, this circRNA was previously reported to be one of the 10 most upregulated circRNA in lung cancer tissue [43].

### 2.4. ML-Classifier Development and Performance for Detection of Lung Cancer Patients Using Human Immunology V2 Panel and 78-circRNA Custom Panel

To evaluate the potential use of the human immunology v2 panel as platelet signature for lung cancer detection, we employed ML approaches (as explained in Materials and Methods). The RFECV algorithm selected a final 28 mRNAs signature (Appendix A and Appendix A). To investigate the performance of different ML algorithms, two ML classifiers were tested (ETC and RF) using 5CV method. RF classifier testing on the 28-mRNA signature leads to the highest ROC AUC of 0.88 ± 0.1 and an accuracy of 76% compared with ETC algorithm. Sensitivity and specificity were respectively 77% and 75%, resulting in 44 out of 58 cases being correctly classified (Figure 5a, Appendix A). Classification scores were significantly different between the lung cancer group and the control group (Mann–Whitney U test *p* < 0.0001, Figure 5b).

The same ML approach was applied to investigate the diagnostic potential of the 78 circRNA custom panel. The RFECV method selected a signature of 21 circRNAs (Appendix A and Appendix A). Both RF and ETC classifiers resulted in a final AUC of 0.81 ± 0.08 and an accuracy of 72% (Figure 5c and Appendix A). The two models differ in sensitivity and specificity; the RF model shows a higher sensitivity (Sensitivity RF: 77%) compared with ECT (Sensitivity ETC: 70%), but a lower specificity (Specificity RF: 67% and Specificity ETC: 73%) (Appendix A). The classification scores of both models were confirmed to be significantly different between the two groups (Mann–Whitney U test *p* < 0.0001, Figure 5d).

### 2.5. Combinatorial Analysis: mRNA and circRNA Signature for the Detection of Lung Cancer Patients

Combinatorial analysis of different types of molecular biomarkers has not yet been investigated in platelets. Our unique cohort of samples allows the exploration of both platelet mRNA and circRNA derived from the same source.

Using the same ML approach applied before, we built a new predictive model using features derived from both the mRNA and circRNA panel (total features = 338) and excluding the two previously identified outlier samples (Appendix A). The RFECV algorithm selected a signature of six mRNAs (BTK, IRAK2, PSMB9, RUNX1, SYK, and LILRB1) and two circRNA (circSLC8A1 and circCHD9) (Appendix A and Appendix A). Once again, the RF classifier yielded the predictive model with the highest ROC AUC (0.92 ± 0.06) and accuracy (81%) (Figure 6a and Appendix A). Sensitivity and specificity were 77% and 87%, respectively (negative predicted value (NPV) = 0.77 and positive predicted value (PPV) = 0.85), resulting in 47 out of 58 samples being correctly classified (Figure 6b). The classification scores of the cancer and control groups showed statistically significant differences (Mann–Whitney U test, *p*-value < 0.0001, Figure 6c).

In terms of AUC, accuracy, and specificity, this model outperforms the results seen in the previous models using an independent signature of mRNA or circRNA, suggesting a potential synergistic role of the combinatorial use of these two RNA types as molecular biomarkers.

### 2.6. Early-Stage Lung Cancer Detection Using Combinatorial Signature of mRNA and circRNA

The outcome of the combinatorial mRNA-circRNA analysis suggests that the inclusion of different RNA types from the same biosource provides a biomarker signature for the detection of lung cancer. Based on these results, we sought to design a computational method for identifying a specific early-stage disease signature. For the identification of this signature, we employed and re-analyzed the 20 early-stage lung cancer samples (stage IA to IIIA) together with the control cohort (*n* = 30) (Appendix A–C). The combinatorial analysis of mRNA and circRNA panel was run through the ML algorithm, which selected a signature of only five features including two circRNAs (circSLC8A1 and circCHD9) and three mRNAs (PSMB9, RUNX1, and LILRB1). Based on this new signature, the algorithm was able to classify early-stage lung cancer samples and controls with an AUC of 0.96 ± 0.03 and an accuracy of 86% (Appendix A). The sensitivity and specificity reached by this early-stage predictive model were 85% and 86%, respectively. Although we observed three false negative samples, which were derived from two patients with stage IIIA and one stage IA (Appendix A), the classification score analysis showed a significant separation of the two groups of interest (Mann–Whitney U test, *p* < 0.0001, Appendix A).

Cumulatively, our data strongly suggest that combinatorial analysis of different RNA types found in blood platelets enables optimal classification of lung cancer patients and demonstrates the potential for early-stage detection.

## 3. Discussion

Platelet transcriptome is a rich source of cancer biomarkers. In this study, we developed a novel and reliable methodology for the interrogation of platelet mRNA and circRNA repertories in order to discover and assess the diagnostic value of each individual RNA type. However, most current liquid biopsy tests rely on the use and analysis of one single type of molecular biomarker, which may often lack the sensitivity and specificity required to obtain clinically reliable information. Therefore, we investigated whether combinatory analysis of platelet mRNA and circRNA derived from the same source may help us to improve the detection of lung cancer patients compared to using the single signature of both types of biomarkers.

Most of the current studies on platelet transcriptome have been based on RNA sequencing data. Although RNA-seq represents a powerful tool to perform high-throughput analysis, its clinical use is limited by the long turnaround time, high cost, and the complex computational analysis. NanoString nCounter technology represents a valid alternative for the clinical implementation of LB tests. Different from the qPCR and NGS assays, this methodology permits a robust and reliable quantification of the RNA molecules without the bias introduced by reverse transcription or amplification. The automated processing minimalizes in-between steps handling errors. The time from sample preparation to data results requires only three days. However, this technology has not yet been largely utilized for liquid biopsy profiling.

Clinical samples, specifically liquid biopsy specimens, often suffer from a limited amount of RNA material for subsequent gene expression analysis. We investigated whether direct usage of platelet RNA in the analysis could provide adequate gene expression profile with the least amount of input. Our findings led us to the conclusion that no pre-amplification step is required to assess gene expression in platelets from as little as 1 ng of total RNA. However, a minimum of 6 ng of RNA is recommended as initial input to reduce intrasample variability and increase the reproducibility of the assay.

In this proof-of-concept study, mRNA and circRNA profiles of human platelets derived from lung cancer patients (*n* = 30) and non-cancer individuals (*n* = 30) were investigated using two different gene panels. The human immunology v2 panel includes 594 genes involved in the immune response such as cytokines, enzymes, interferons, and their receptors [37]. Out of the 594 mRNAs present in the panel, 435 mRNAs were detected in platelet samples analyzed, whereas 18 were exclusively expressed in the control group and 15 in the cancer samples. The second custom-made panel comprised 78 circRNA targets, including circRNA candidates described to be differentially expressed in lung cancer tissues, cell lines, or body fluids [38]. All 78 targets were detected in platelet samples investigated. Three of them appear to be present exclusively in the cancer group. These three circRNAs were previously found dysregulated in lung cancer tissues with an important role in cancer progression and regulation [39,44,45]. They function as a sponge and regulate the activity of important miRNA, controlling tumorigenesis, cancer progression, and proliferation processes [39,40,41,42].

In order to analyze and determine the diagnostic potential of platelet transcriptome, we developed a complete computational workflow based on nCounter data analysis and machine learning. This bioinformatic pipeline can be divided essentially into four main parts (Figure 2).

In the first part, the quality controls and the filtering of possible sample and gene outliers are performed. This step is particularly important to improve and correct the data to obtain an optimal normalization and reduce bias due to the intra-variability of the samples. Based on these criteria, only two samples processed with human immunology v2 panel were excluded from downstream analysis (Control-3 and Control-5).

In the second and third parts, we used and assessed two different biostatistical packages for normalization and DE analysis. Based on RLE plot analysis, DESeq2 outperformed edgeR normalization for both panels studied. DE analysis of the mRNA panel resulted in a total of 25 DE mRNA (Figure 4b). According to gene ontology (GO) analysis, the upregulated genes are mostly involved in inflammatory pathways mediated by chemokine and cytokine signaling, oxidative stress response, and cell signaling. While the downregulated genes are mainly associated with B cell and T cell activation, EGF, TGFβ, Wnt, PDGF signaling pathway, and inflammatory response. The circRNA DE analysis indicates only one significant differentially expressed circRNA between the cancer and control group (Figure 4d). Previous studies confirmed hsa_circRNA_101367 (circFUT8) as one of the most upregulated circRNA in lung cancer [43]. This circRNA can regulate the proliferation, invasion, and apoptosis of lung cancer cells by sponging miR-145 or controlling miR-944/YES1 axis [46,47].

The fourth section of this dry lab workflow employs machine learning approaches to generate prediction models. ML can be considered a novel method for developing predictive signatures that typically outperforms individual biomarkers identified by differential expression analysis.

Using individual mRNA and circRNA data profiles, the ML prediction models generated reached an AUC of 0.88 using a selected signature of 28-mRNA and an AUC of 0.81 using a 21-circRNA signature (Figure 5a,c). However, the combinatorial analysis performed by combined data derived from both RNA types outperforms the results obtained with the single signature. The RFECV algorithm identified a signature of only eight biomarkers (six mRNA and two circRNA), six of which (BTK, PSMB9, RUNX1, SYK, LILRB1, and circSLC8A1) were previously selected in the individual mRNA and circRNA signatures, while IRAK2 and circCHD9 were newly included. Using these features, the prediction model showed an AUC of 0.92 with a sensitivity of 77% and a specificity of 87% using the RF classifier (Figure 6a). Combinatorial analysis not only reduces the number of features of the predictive model, but it also increases the AUC, improving the classification of the two groups of interest. These results indicate that a combination of different types of biomarkers possibly enhances the prediction value over that of single ones.

Despite improvements in terms of AUC, accuracy, and specificity, an increase in the sensitivity of the test is not observed. Post-analysis examination of incorrectly classified samples indicated that six out of the seven false negative samples originated from patients diagnosed with stage III (*n* = 3) and stage IV (*n* = 3). This implies that the selected biomarkers from our prediction model most likely reflect the gene expression signature of the earlier stages of the disease. This hypothesis was further supported by the combinatorial analysis performed only with samples diagnosed as surgically resectable tumors (stages Ia–IIIa). This model, indeed, confirmed that five out of the eight biomarkers previously selected (circSLC8A1, circCHD9, PSMB9, RUNX1, and LILRB1) generated a predictive model specifically for early-stage cancer detection reaching an AUC of 0.96, sensitivity of 85%, and specificity of 86% (Appendix A). Taken together, current findings suggest that these biomarkers may be sensitive to detecting lung cancer at early stage.

Although the restricted number of platelet samples used in our current study imposes a limitation, our proof-of-concept results seem encouraging. This also includes the results from a small group of individuals diagnosed with lung nodules, as a control for non-cancerous disease, that were correctly classified by all our prediction models. A larger cohort of samples for the training and an independent validation group is needed to confirm the clinical efficacy of the combinatorial mRNA-circRNA signatures identified.

Platelet transcriptome is a promising liquid biopsy biosource of cancer-related biomarkers. Although the methodology for generating platelets-derived transcriptome analysis is available [21], the implementation of platelet-derived tests in routine practice is currently hampered by a lack of standardized automated procedures for collecting and processing large numbers of clinical samples in multicenter settings and clinical validation. In this study, our goal was to design and establish, for the first time, a workflow for the nCounter analysis of mRNA and circRNA from platelets for the development of a liquid biopsy test for the detection of lung cancer. We have demonstrated the feasibility of using nCounter for the investigation of both platelet-derived mRNAs and circRNAs, including differential expression analysis, and the development of an ML predictive model. Importantly, our results, using a first multi-analytical approach for combinatorial analysis of mRNA and circRNA signature derived from blood platelets, emphasizes that the combination of the different types of RNAs may help to improve the detection of early-stage lung cancer patients.

## 4. Materials and Methods

### 4.1. Sample Collection and Population Study

Whole blood samples from lung cancer patients (*n* = 30), asymptomatic individuals (*n* = 27) and people with benign lung nodules (*n* = 3) were provided by the Amsterdam UMC (VU University Medical Center, Amsterdam, The Netherlands) and Maastricht University Medical Center (Maastricht, The Netherlands). Whole blood was drawn at the Amsterdam UMC into EDTA-coated BD Vacutainer tubes. At the Maastricht University Medical Center, BD Vacutainer tubes containing 3.2% buffered sodium citrate were used for blood-sample collection. Both collection protocols guarantee minimal platelet activation [10,21,48,49].

Patients with cancer had their blood drawn at the time of diagnosis or, in the event of surgically treatable (resectable) tumors, one day before surgery. Histological analysis of the tumor tissue biopsy was performed to determine the diagnosis. Asymptomatic individuals had no prior or current medical records of any kind of cancer during the time of blood collection and no additional examinations were carried out to verify the absence of cancer.

Clinical information about the patients was gathered, including their age, gender, type of tumor, and level of metastasis (Appendix A). For the current study, age- and gender-matching was done by incorporating samples of non-cancer controls and cancer patients with comparable median ages and gender distributions between the two groups.

Clinical follow-up of asymptomatic controls was not available due to the anonymization of these samples in accordance with the ethical guidelines of the hospitals. The Declaration of Helsinki’s guiding principles were followed in the conduct of this investigation. This study has received approval from the medical ethics committees of the two participating hospitals (approval code: 11-4-117.4/pl, 2016.268 and 2017.545). The informed permission form for blood collection and blood platelet analysis was given to and signed by each participant.

### 4.2. Isolation of Blood Platelets

Platelets isolation from the whole blood sample was performed as previously described [21]. Briefly, to separate platelet-rich plasma (PRP) and nucleated blood cells, collected blood was spun at 120× *g* for 20 min, followed by PRP centrifugation at 360× *g* for 20 min at room temperature. Resulting platelets pellet was re-suspended in RNAlater (Thermo Scientific, Waltham, MA, USA), incubated at 4 °C over-night, and stored at −80 °C until use.

At the Maastricht University Medical Center, PRP was obtained by centrifuging blood sample at 240× *g* for 15 min. PRP was supplemented with iloprost (50 nM) to reduce ex vivo platelet activation. PRP was centrifuged for two minutes at 1600× *g* to pellet the platelets, followed by the addition of RNAlater and storage at −80 °C until use. Both procedures guarantee the isolation of highly pure platelet pellets with minimal leukocyte contamination and platelet activation. There were no discernible deviations detected in downstream analyses between the two methods [21,48,49].

### 4.3. Total RNA Isolation

Total RNA isolation was carried out using the mirVana RNA isolation kit according to the manufacturer’s instructions (Ambion, Thermo Scientific, cat. no. AM1560). Extracted RNA was eluded in 30 μL of mirVana buffer and the quantity and quality were assessed by RNA 6000 Picochip (Bioanalyzer 2100, Agilent, Santa Clara, CA, USA). RNA samples with RIN values higher than 7 and/or with distinguishable rRNA peaks were considered for further analysis.

### 4.4. Gene Expression Analysis Using nCounter

The assays were performed using the NanoString nCounter Flex System (NanoString Technologies, Seattle, WA, USA) with two different nCounter panels for the analysis of platelet-derived RNA. The human immunology v2 panel (NanoString Technologies) targets 594 genes involved in the immune response such as cytokines, enzymes, interferons, and their receptors [37]. For each sample, 6 ng of total platelet RNA was hybridized with the biotinylated capture probe and the reporter probe attached to color-barcode tags for 18 h at 65 °C. The second panel was a custom-made panel targeting 78 circRNAs (78-circRNA panel), 6 linear reference genes and 4 mRNAs [38]. For this analysis, 8 ng of total platelet RNA from each sample was hybridized with the capture and reporter probes for 18 h at 67 °C.

The automated nCounter^®^ Prep Station was used to process the samples. The samples were purified and immobilized in a sample cartridge for data collection, where the target mRNA and circRNA in each hybridized sample were quantified, using the nCounter^®^ Digital Analyzer. Output data in the report code count (RCC) format was exported into the nSolver analysis software (version 4.0.70). The background of each sample was computed using the geomean of the counts of the negative probe (negative controls, NCs) plus two times the standard deviation. Raw counts below the negative background value were excluded from further analysis.

### 4.5. Data Normalization and Differential Expression Analysis

Pre-processing and normalization of the data were performed using R (version 4.0.3) and RStudio as graphical interface (version 2022.02.2). The quality of the raw RCC proprietary format data was initially assessed by using the NanoStringQCPro (version 1.22.0) package. Standard control metrics embedded by NanoString, such as imaging, binding density, positive control linearity, and limit of detection, were used to search for any potential outlier samples.

Additionally, all samples were also subjected to supplementary exploratory examination, including the principal component analysis (PCA) and inter quartile range (IQR) method for outlier detection. Samples higher than the upper bound (Q3 + 1.5 × IQR) or lower than the lower bound (Q1 − 1.5 × IQR) were excluded from subsequent analysis.

Prior to normalization, negative control probes embedded to each panel were used to filter out targets with poor expression and high background noise. Consequently, the background values were firstly calculated, by taking the mean of each sample’s negative controls increased by two times the standard deviation, and then removed from each sample. Any transcript that indicated a score of less or equal to 0 in more than 75% of the examined samples was excluded from further examination. After these filtering steps, the data was again evaluated using a PCA plot. Two different packages were compared for the normalization of the data: DESeq2 (version 1.30.1) and edgeR (version 3.32.1). The normalization performance was assessed using the standard relative log expression (RLE) plot. DESeq2 was chosen as the default to perform the normalization of the data. Differential expression (DE) analysis was performed to find significantly differentially (|FC| > 0.5 and *p*-adj < 0.05) expressed genes between the cancer and control groups.

### 4.6. Feature Selection and Classification Analysis

The machine learning approach was implemented in Python (v3.9.13) using the Scikit-learn (v1.1.0) library. Initially, the DESeq2-normalised data, along with each sample’s classification label, were imported into the python environment. For combinatorial analysis, the mRNA and circRNA normalized datasets were merged together with previous analysis. Highly correlated (higher than 0.95), as well as quasi-constant features, were excluded from further analysis.

The recursive feature elimination with cross-validation (RFECV) algorithm was then utilized along with the random forest (RF) classifier to perform the feature selection in addition to the leave-one-out cross-validator (LOOCV). RFECV determined automatically the number and the composition of the most relevant features. This subset of genes, which composes the prognostic gene signature, would further be used as an input to our classification models.

Two different supervised machine learning algorithms, RF and extra trees classifiers (ETC), were selected along with the selected features to perform this classification problem. In our case, the 5-fold cross-validation (5CV) was used. In a more detailed manner, the dataset was randomly divided into 5 folds, with 4/5 of the data being used to train the model and the remaining 1/5 being used to test its behavior. This process was repeated 5 times. The use of k = 5 was chosen to reduce the bias in the testing set due to the limited number of samples available. The classifier with the highest mean AUC ROC value was then selected. Probability scores for each sample were obtained from the final classifier. Finally, additional statistical metrics such as sensitivity, specificity, accuracy, PPV, and NPV were also calculated.

## Figures and Tables

**Figure 1 ijms-24-04881-f001:**
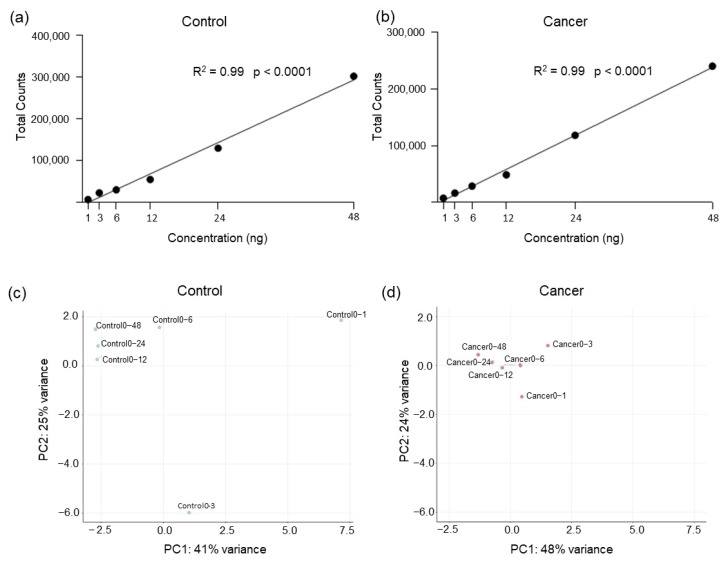
Titration experiment using total RNA derived from blood platelets of cancer patient and asymptomatic individual. (**a**) Six different inputs (1 ng, 3 ng, 6 ng, 12 ng, 24 ng and 48 ng) of total RNA derived from platelets of a non-cancer individual (Control) were tested using human immunology v2 panel with nCounter platform. Total number of counts detected (after negative background removal) follow a linear regression model (R^2^ = 0.99, *p*-value < 0.0001). (**b**) A similar experiment was performed using total RNA derived from lung cancer platelet sample, and in this case, the total number of counts after background removal follow a linear regression model (R^2^ = 0.99, *p*-value < 0.0001). (**c**) Principal Component Analysis (PCA) assessing RNA profile of Control sample using six different initial inputs of total RNA. (**d**) PCA assessing RNA profile of Cancer sample using six different initial inputs of total RNA.

**Figure 2 ijms-24-04881-f002:**
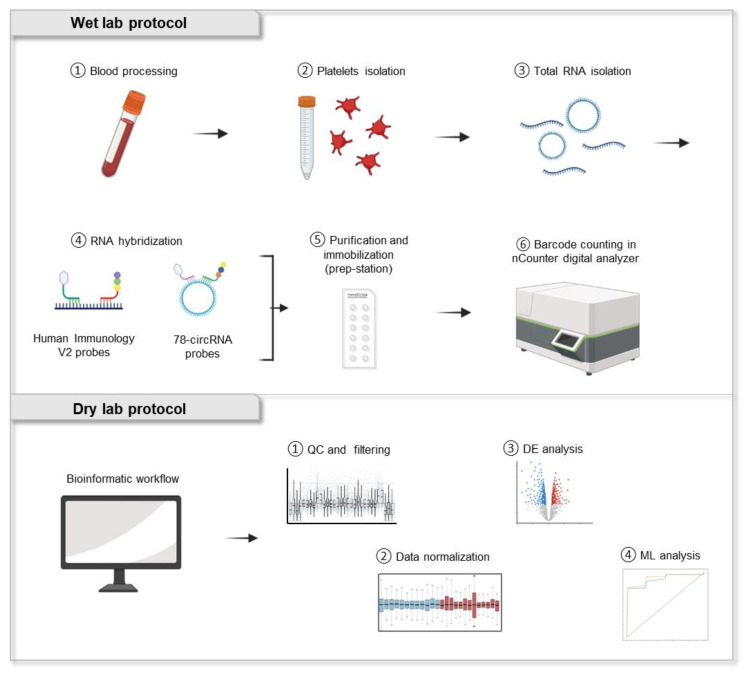
Wet and dry lab workflow for the study of mRNA and circRNA derived from blood platelets using nCounter technology. Total RNA is extracted from human platelets and directly hybridized with nCounter probes. Hybridization and purification are performed on the nCounter prep-station and counting of the hybridized barcode is performed on the nCounter digital analyzer. Bioinformatic workflow consists of quality control (QC) and data filtering, normalization of counts, differential expression (DE) analysis, and the generation of a prediction model through the use of ML algorithms.

**Figure 3 ijms-24-04881-f003:**
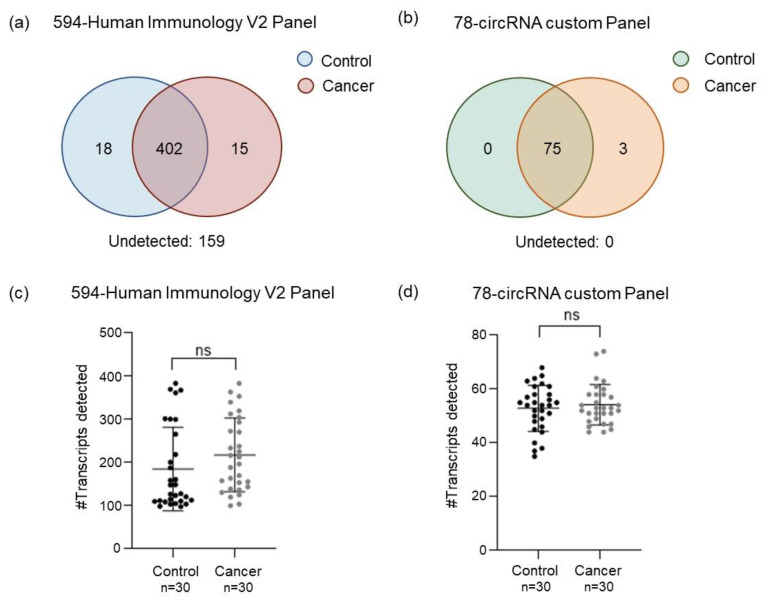
Platelet-mRNA and circRNA detection using human immunology v2 panel and 78-circRNA panel. (**a**) Venn diagram showing mRNAs identified in lung cancer and control samples using human immunology v2 panel. (**b**) Venn diagram showing circRNAs identified in lung cancer and control samples using 78-circRNA custom panel. (**c**) Number of transcripts detected in blood platelets derived from cancer patients and non-cancer controls using human immunology v2 panel (Mann–Whitney U test, *p*-value > 0.05). (**d**) Number of circRNAs detected in blood platelets derived from cancer patients and non-cancer controls using 78-circRNA custom panel (Mann–Whitney U test, *p*-value > 0.05).

**Figure 4 ijms-24-04881-f004:**
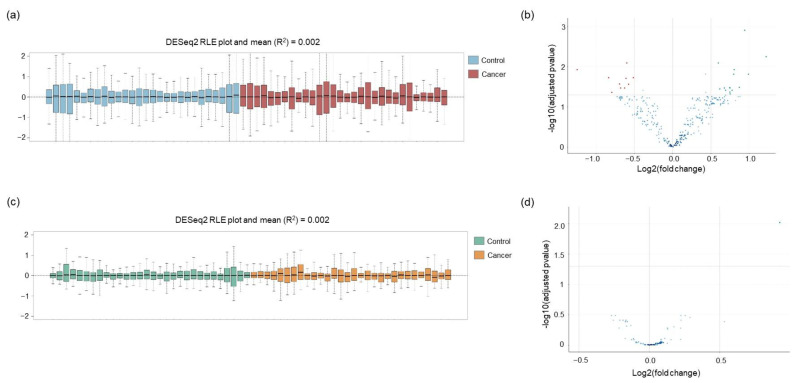
Normalization and differential expression analysis of mRNA and circRNA. (**a**) RLE plot of the normalized mRNA data generated using DESeq2. (**b**) Volcano plot of differentially expressed mRNAs. The negative log of the adjusted *p*-value (base 10) is plotted on the Y-axis, and the log of the FC (base 2) is plotted on the *X*-axis. Red dots indicate significantly downregulated mRNA and green dots represent significantly upregulated mRNA (adjusted *p*-value < 0.05). (**c**) RLE plot of the normalized circRNA data generated using DESeq2. (**d**) Volcano plot of differentially expressed circRNAs. Green dot represents the significantly upregulated circRNA (adjusted *p*-value < 0.05).

**Figure 5 ijms-24-04881-f005:**
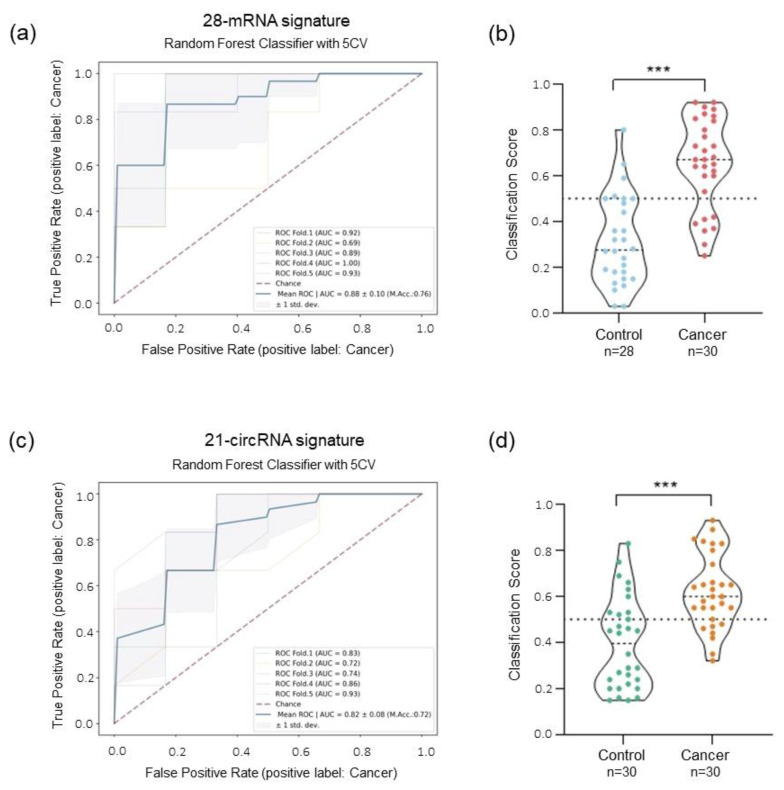
ML analysis using single signature of mRNA and circRNA. (**a**) AUC ROC curve of the 28-mRNA signature using RF classifier for group classification. (**b**) Violin plot of the classification score of samples generated using the 28-mRNA predictive model (*** indicates *p*-value < 0.001 in a two-tailed Mann–Whitney U test). (**c**) AUC ROC curve of the 21-circRNA signature using RF classifier for group classification. (**d**) Violin plot of the classification score of samples generated using the 21-circRNA predictive model (*** indicates *p*-value < 0.001 in a two-tailed Mann–Whitney U test).

**Figure 6 ijms-24-04881-f006:**
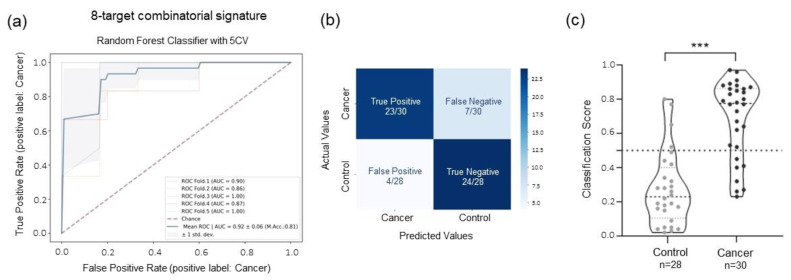
ML analysis using combinatorial signature of mRNA and circRNA. (**a**) AUC ROC curve of the 8-mRNA-circRNA signature using RF classifier for group classification. (**b**) Confusion matrix indicating the number of the correctly classified and misclassified samples based on the previous prediction model. (**c**) Violin plot of the classification score per samples generated using the 8-mRNA-circRNA predictive model (*** indicates *p*-value < 0.001 in a two-tailored Mann–Whitney U test).

**Table 1 ijms-24-04881-t001:** Data table with the general characteristics of sample cohort (non-cancer individuals and lung cancer patients) including the number of samples, age, sex, and clinical information employed for the analysis with both human immunology v2 and 78-circRNA custom panel using nCounter technology.

	Control	Cancer
No. of samples	30	30
Age (average; min–max)	63.1 (42–79)	63.3 (51–79)
Female	16	15
Male	14	15
Lung nodules	3	-
Early-stage (stage I to IIIa)	-	20
Late-stage (stage IV)	-	10

## Data Availability

The raw nCounter NanoString data RCC-files generated and analyzed during the current study are available in the NCBI GEO database under accession number GSE225787.

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
