# Peer review of "Combinatorial Blood Platelets-Derived circRNA and mRNA Signature for Early-Stage Lung Cancer Detection"

_ijms, 2023, doi:10.3390/ijms24054881_

Round 1
Reviewer 1 Report
Per attached file

Reviewer 2 Report
Thank you for the chance you gave me to read this interesting study entitled “Combinatorial blood platelets-derived circRNA and mRNA 2 signature for early-stage lung cancer detection” by Silvia D’Ambrosi et al. In this original research paper, the authors investigated the synergistic empowerment of circRNA and mRNA signatures derived from blood platelets as biomarkers for lung cancer detection. They developed a protocol exploiting platelet mRNA and circRNA repertoire by using Nanostring nCounter technology and machine learning approaches. Through a multi-analyte-based approach, the authors improved their initial findings resulting in an 8-target signature (6 40 mRNA and 2 circRNA) and a very high AUC (0.92). This topic is clinically oriented and has great importance. The study is well-designed and organized as well as well presented. I think that this study in the current form does satisfy the appropriate criteria for publication in your journal, however, some minor points should be treated before being suitable for publication.
Some minor points are:
Although the score (20%) obtained by the plagiarism detection service “Turnitin” is low, the authors could improve this issue especially focusing on the Methods section.
Although the manuscript is generally well written, however, it needs to be literally edited for some minor mistakes and strange structures.
Introduction
Lines 52-54: Please, double check this sentence since the percentage of 7% sounds very low and it is not included in the mentioned study.
Methods
The authors should discuss whether the different anticoagulants used during the blood collection in the two medical centers could influence the results and if they have tested this. In this vein, the different platelets isolation methodologies could also influence the results?
Lines 457-459: This statement should be supported either by published data or by new experimental work.
The authors should provide the decision number of ethics committees of the two participating hospitals.
Discussion
Limitation paragraph should be improved including some methodological issues (blood collection, different isolation methodologies etc).
Round 2
Author Response
We appreciate the reviewer's feedback. We took their suggestions into account and made the necessary changes to the text.
